

# Only the instantaneous global warming potential is consistent with honest and responsible greenhouse gas accounting

Peter Nightingale

Department of Physics, University of Rhode Island, Kingston, Rhode Island 02881, U.S.A.

**Correspondence:** Peter Nightingale (nightingale@uri.edu)

**Abstract.**

This paper presents a simple model to describe the impact on global warming of methane (natural gas) when used for energy production. The model is used to estimate the near-term effect of energy policies based on natural gas as a bridge fuel. The results make it clear that the commonly employed global warming potential of methane with a 100-year time horizon has the

following problems:

**1:** it produces misleading results;

**2:** is inconsistent with meaningful tracking of greenhouse gas emissions; and

**3:** is incompatible with the precautionary principle.

## 1   Introduction

In 2008, Hansen et al. (2008) argued that atmospheric $CO_2$ concentration exceeding 350 ppm poses an unacceptable danger to human existence. As the authors put it in their abstract:

> If humanity wishes to preserve a planet similar to that on which civilization developed and to which life on Earth
> is adapted, paleoclimate evidence and ongoing climate change suggest that $CO_2$ will need to be reduced from its
> current 385 ppm to at most 350 ppm, but likely less than that. The largest uncertainty in the target arises from
> possible changes of non-$CO_2$ forcings.

The paper warned that continued growth of greenhouse gas emissions for just another decade—after 2008—would make it practically impossible to avoid catastrophic effects on the climate system. Nonetheless, a decade later, atmospheric $CO_2$ is fluctuating around 410 ppm (Pro Oxygen, 2018) and it is still increasing at a rate of 2–2.5 percent per year.

In the aforementioned quote, the authors qualify the critical number, 350 ppm, mentioning that it is likely too high, as it

fails to account fully for the dangers of non-$CO_2$ forcings. Of these, atmospheric methane is the dominant one. What is more, methane forcing is currently far more dangerous than it was in 2008. For instance, Turner et al. (2016) concluded on the basis satellite and surface data that there had been a large increase in the methane emissions of the United States over the decade prior to their 2016 study. Worden et al. (2017) subsequently traced these increases back to fossil fuel sources.





At the same time, it has become increasingly clear that the climate system is deteriorating at the rate outpacing projections, those of the Intergovernmental Panel on Climate Change (IPCC) in particular. Brysse *et al.* discuss numerous examples of scientists "erring on the side of least drama" (Brysse et al., 2013). Among the most disconcerting such under-predicted devel-opments is the Arctic amplification documented in recent Arctic Report Cards issued by the United States National Oceanic
Atmospheric Administration (National Oceanic Atmospheric Administration, a, b).

Because of the shift in the United States to natural gas,[1] which is used increasingly for the generation of electricity, it is imperative to reconsider the overemphasis in policy decisions on $CO_2$ emissions. This overemphasis results from the common practice of using the 100-year horizon global warming potential in the calculation of the $CO_2$ equivalents upon which climate policies are based persuant to the United Nations Framework Convention on Climate Change (United Nations Framework
Convention on Climate Change (UNFCCC), 2008).

The *Fifth Assessment Report* of Intergovernmental Panel on Climate Change (IPCC) explicitly states that there is no scientific argument for using the 100-year horizon—see, *e.g.,* page 711 of Myhre et al. (2013). To the contrary, when the time scale of climate change is likely to be decadal, it is vital *not* to use the 100-year horizon as the basis of major energy policy decisions. Stated more bluntly, given the danger to which it subjects life on Earth, this choice is irresponsible and constitutes a patent
violation of the precautionary principle, number 15 of the Rio Declaration Assembly (1992); Change (1994).

This paper presents a simple dynamical model that produces order of magnitude estimates using the instantaneous global warming potential rather than the one based on the 100-year horizon. The results form the basis of the statement made above global warming potentials based on 100-year time horizon must be abandoned once and for all for major policy development. (Edwards and Trancik, 2014) presented a similar line of reasoning, one that focusses on a dynamical approach rather than the
static one that is implicit in the use of any non-instantaneous global warming potential.

The layout of the paper is as follows: in Section 2 reviews some of the well-known basic properties of methane and introduces the simple dynamical model mentioned before. Section 3 presents results of some simple energy policy thought experiments. Finally, Section 4 summarizes the conclusions.

## 2   Methane basics

To estimate the effect of a given greenhouse gas on global warming one uses the global warming potential mentioned in Sec-tion 1. This quantity is a dimensionless multiplier that converts the effect of the emission of a unit mass pulse of a greenhouse gas under consideration to a mass of $CO_2$ that would have the same global warming effect, the $CO_2$ equivalent ($CO_2$e) mass A pulse of $CO_2$ injected into the atmosphere is taken up by the ocean, biosphere and soil and decays by half in about 25 years but 20% is still in the atmosphere after 500 years; see Fig. 4A in Hansen et al. (2013).
Atmospheric $CH_4$ has a half-life of less than a decade. The global warming potential, as defined in Section 8.7.1.2 of Myhre et al. (2013), is a fraction: the time-integral of the radiative forcing due to a pulse emission of a given greenhouse gas divided

---

[1]Exploration and production subsidies from the federal government have increased dramtically, during the Obama administration; see, *e.g,* Oilchange International. This trend is exptected to continue or accelarate during the Trump administration.





by same quantity for a pulse of an equal mass of $CO_2$. Due to the atmospheric dynamics of both $CO_2$ and $CH_4$ the resulting global warming potential depends on the time interval used in the integrals, aka the time horizon. The global warming potential is denoted by $G_t$ with $t$ the time horizon measured in years. Use of this quantity to make predictions about the climate system anounts to an uncotrolled approxiation, but one would expect it to yield reasonoble order of magnitude estimates if used with

care.

Unfortunately, however, the choice of the itime horizon $t$ is a major source of confusion. In addition to the value judgment mentioned in Section 1 and acknowledged by the Intergovernmental Panel on Climate Change (IPCC), there is an issue of the time scale relevant to the process or policy decision under consideration. For a project small on a global scale, averaging emissions over the expected life time of that project makes physical sense, but for matters of global scale, such as the energy

policy of the United States, the horizon should be set by the time scale of the climate change phenomena and the danger they pose to life on Earth. Therefore, one has to consider the following:

1. The arguments made by Hansen in Hansen (2005) and the well-known difficulty of predicting instabilities (aka state shifts or tipping points) such as the sudden and, on a human multi-generational time scale irreversible, disintegration of ice sheets.

2. Recent developments on a decadal time scale in the Arctic (National Oceanic Atmospheric Administration, a; **?**).

3. The obligation to heed the precautionary principle of Rio Declaration mentioned in the Section 1 (Assembly, 1992; Change, 1994).

Based on these physical considerations, and the simple, mathematical fact that non-instantaneous global warming potentials are incompatible with a dynamical approach, this paper uses the instantaneous global warming potential $G_0$.

The effect of the choice of the horizon manifests itself in the critical fraction $f_c$ of fugitive $CH_4$ above which the global warming impact of the unburned, fugitive methane cancels out its higher energy density per unit emitted $CO_2$. To find $f_c$, suppose one generates energy from one mole of $CH_4$ a fraction $f$ of which escapes unburned. The part that is burned adds $(1-f)$ moles of $CO_2$ to the atmosphere. Given $G_t$, the global warming potential of $CH_4$, the fraction $f$ of fugitive $CH_4$ adds $(4/11)fG_t \equiv G'_t$ to the atmospheric $CO_2$ equivalent. The total increase is $1 - f + fG'_t$. Note that the molecular mass ratio

$4/11$ of $CH_4$ and $CO_2$ appears because of the conventional definition of the global warming potential $G_t$ which compares the effects of a *mass* of $CH_4$ of to the effect of the same *mass* of $CO_2$, rather than the same number of moles; see (Myhre et al., 2013, p. 710).

Different fuels emit different amounts of $CO_2$ per unit energy produced upon combustion. Suppose that per unit $CO_2$ produced, $CH_4$ generates a factor $\varepsilon$ more energy than some other fuel, say coal or oil. For coal the calculations in this paper

use $\varepsilon = 2$ and for oil $\varepsilon = 4/3$ (U.S. Energy Information Administration (EIA)). Taking into account the fugitive gas loss of $CH_4$, to produce the same amount of electric energy as from $CH_4$, one has to burn a relative amount of $(1-f)\varepsilon$ coal or oil.

The critical fraction $f_c$ for which both processes have the same impact on the climate follows from the equation

$$1 - f_c + f_c G'_t = (1 - f_c)\varepsilon, \tag{1}$$



**Table 1.** Critical fractions $f_c$ for coal and oil for global warming potentials $G_t$ with various time horizons $t$ in units of years.

|  | $G_0 = 120$ | $G_{20} = 34$ | $G_{100} = 86$ |
|---|---|---|---|
| $\varepsilon = 2$ (coal) | 2.2% | 3.1% | 7.5% |
| $\varepsilon = \frac{4}{3}$ (oil) | 0.76% | 1.1% | 2.6% |

so that

$$f_c = \frac{\varepsilon - 1}{\varepsilon - 1 + G'_t}. \tag{2}$$

Tab. 1 shows the critical fractions for fugitive $CH_4$ for various time horizons and fuels.[2]

Before discussing the relevant kinetic equations, we recall that the solution of the decay equation with decay time $\tau$ for any

$g(t)$ with source $s(t)$ subject to initial condition $g(0) = 0$,

$$\dot{g}(t) = -g(t)/\tau + s(t), \tag{3}$$

where $\dot{g} = dg/dt$, is given by

$$g(t) = \int_0^t e^{-(t-t')/\tau} s(t') \, dt', \tag{4}$$

for $t \geq 0$.

For the kinetic equations it is convenient to use molar number densities: $c(t)$ for $CO_2$, $m(t)$ for $CH_4$, and $c_e(t)$ for $CO_2$ equivalent of the mix. Generalization is straightforward, but to simplify the thought experiment presented in this paper and obtain the order-of-magnitude estimates of interest—the results of which are in Section 3—it suffices to account only for the greenhouse gases $CO_2$ and $CH_4$. The $CO_2$ equivalent is given by:

$$c_e(t) = c(t) + G'_0 m(t). \tag{5}$$

A further assumption in this thought experiment is that all of the increase in atmospheric $CO_2$ comes from the hypothetical future use of methane only. Then there are the following sources for increased emissions: *(i)* the combustion of $CH_4$; and *(ii)* the oxidation of fugitive methane as it decays in the atmosphere.

That is, if $p(t)$ is the rate of increase in $CO_2$ produced by coal or oil, using methane to generate the same power, yields the following rate of increase of $CO_2$:

$$\dot{c}(t) = p(t)/\varepsilon + m(t)/\tau, \tag{6}$$

where the last term arises from the $CO_2$ production rate due to the oxidation of atmospheric $CH_4$; here $\tau = 12.4\,\text{year}$, the atmospheric decay time of $CH_4$ ((Myhre et al., 2013, Table 8.7)). The rate of increase of $CH_4$ is:

$$\dot{m}(t) = -m(t)/\tau + \frac{f}{(1-f)\varepsilon} p(t), \tag{7}$$

---

[2]For the global warming potential $G_t$ see (Myhre et al., 2013, Table 8.7), which contains the numbers for the 20- and 100-year horizons, *viz.* 86 and 34. For further details and the instantaneous global warming potential see Fig. 8.29, Tables 8.7 and 8.A.1 on pages 712, 714, and 731, *ibid.*


the last term accounts for the emission of fugitive $CH_4$. The desired solution of the differential equations corresponds to the hypothetical case in which for $t < 0$ power generated by combustion of coal and oil only. At $t = 0$ the a complete switch takes place to $CH_4$. The corresponding solution, subject to initial condition $m(0) = 0$, is

$$m(t) = \int_0^t e^{-(t-t')/\tau} \frac{f}{(1-f)\varepsilon} p(t_1)\, dt_1. \tag{8}$$

Substitute Eq. (8) into Eq. (6) and integrate, assuming $c(-\infty) = 0$

$$c(t) = \int_{-\infty}^0 p(t_1)\, dt_1 + \int_0^t [p(t_1)/\varepsilon + m(t_1)/\tau]\, dt_1. \tag{9}$$

The final result for the atmospheric $CO_2$ equivalent concentration is obtained by substituting Eqs. (8) and (9) into Eq. (5). The result is:

$$c_\mathrm{e} = \int_{-\infty}^0 p(t)\, dt + \frac{1}{\varepsilon}\int_0^t p(t_1)\, dt_1 + \frac{f}{(1-f)\varepsilon}\left[ G_0' \int_0^t e^{\frac{t_1-t}{\tau}} p(t_1)\, dt_1 + \frac{1}{\tau}\int_0^t \int_0^{t_1} e^{\frac{t_2-t_1}{\tau}} p(t_2)\, dt_2\, dt_1 \right]. \tag{10}$$

In the case discussed in this paper, the function $p$ is represented accurately by a simple exponential, as shown in the next Section 3, so that the integrals can be done exactly; in more complicated cases, numerical integration is straightforward. In practical applications of a dynamical scheme of this sort it would suffice to use a finite-difference approximation based on yearly data and appropriately chosen initial conditions.

## 3 Results

Estimates of total carbon dioxide emissions from the beginning of the Industrial Revolution are available from CDIAC (Carbon Dioxide Information Analysis Center (CDIAC), 2014). As shown in Fig. 1, the data can be represented surprisingly accurately by a simple exponential growth curve; the curve shown in Fig. 1 satisfies the equation

$$C_\mathrm{global}(t) = 9.00\, e^{0.025\left(\frac{t}{\mathrm{year}} - 2010\right)} \text{ GtC/year}. \tag{11}$$

This equation is used to define "continuing business as usual." The expression was obtained by a least squares fit, followed by a slight adjustment of the normalization constant so that the integral from $-\infty$ to year 2011 reproduces

$$\int_{-\infty}^{2011\,\mathrm{year}} C_\mathrm{global}(t)\, dt = 365\,\mathrm{GtC}, \tag{12}$$

the CDIAC estimate of 2011 cumulative emissions (Carbon Dioxide Information Analysis Center (CDIAC), 2014).

Given that $CO_2$ emissions are the predominant driver of global warming, it is not surprising that temperature anomaly

$T$, shown in Fig. 2, is consistent with the climate forcing resulting from these emissions. The temperature anomaly data of





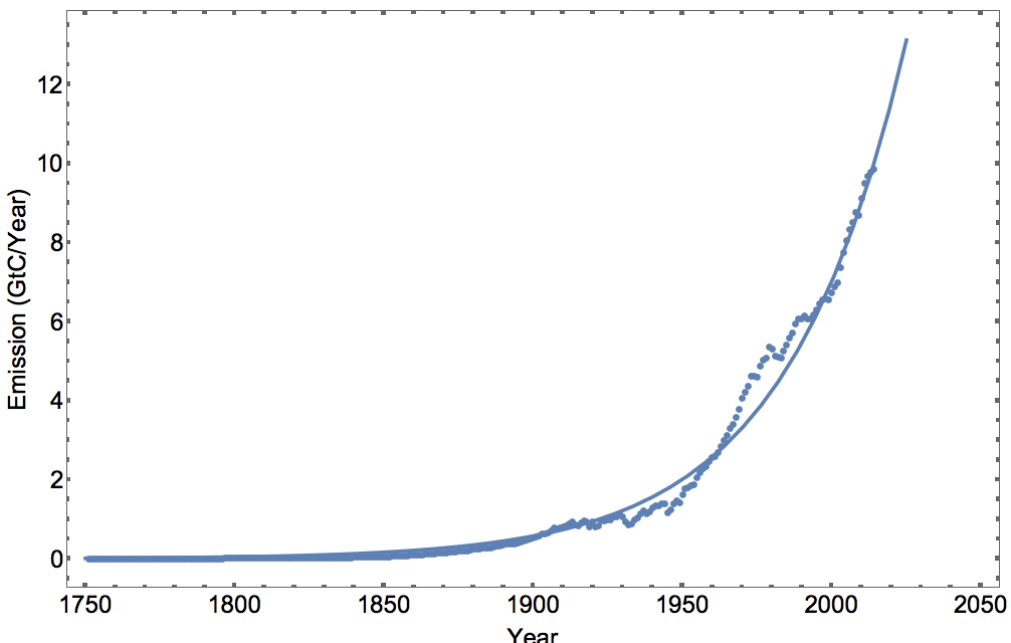

**Figure 1.** Global $CO_2$ emissions in gigatons of carbon per year with exponential fit

NASA/GISS (NASA, 2017) can be used for a linear regression, two-parameter least-squares fit using the same exponential function used in Eq. (11). This yields the following expression

$$T(t) = -0.3\,°\text{C} + 1.0\,°\text{C}\,e^{0.025\left(\frac{t}{\text{year}} - 2010\right)}, \tag{13}$$

shown as the solid curve in Fig. 2.

Here are the results of one thought experiment: assume, first of all, that business-as-usual continues and that global energy consumption keeps growing exponentially, and, secondly, that power is generated by combustion of coal or oil before 2018 and of $CH_4$ after that.

    This produces Fig. 3 in which the solid black curve on the left represents the actual, historical development, a trajectory continued on the right. The blue curve starting in 2018 corresponds to a hypothetical, complete switch to $CH_4$ in that year

with 6% of the $CH_4$ escaping unburned, *i.e.,* half of Howarth's estimate (Howarth, 2015). The red curve corresponds to 12% fugitive $CO_2$. Fig. 4 is the same assuming that combustion of oil generates power before 2018. Because the efficiency increase is considerably less in this case, the deleterious effect of the fugitive $CH_4$ is more pronounced in this case.

    Of course real life is not quite as simple as this thought experiment. However that may be, the results clearly show that, although the red and blue $CH_4$ curves will ultimately cross the black coal or oil curves, this does not happens sufficiently

rapidly to justify the role of $CH_4$, purported as a bridge fuel.

    Business-as-usual is one pathway, another one is to stay within the carbon budget proposed by Hansen et al. (2013). The laws of nature allow humanity not to overspend this budget, but the required replacement of fossil fuels by renewables and



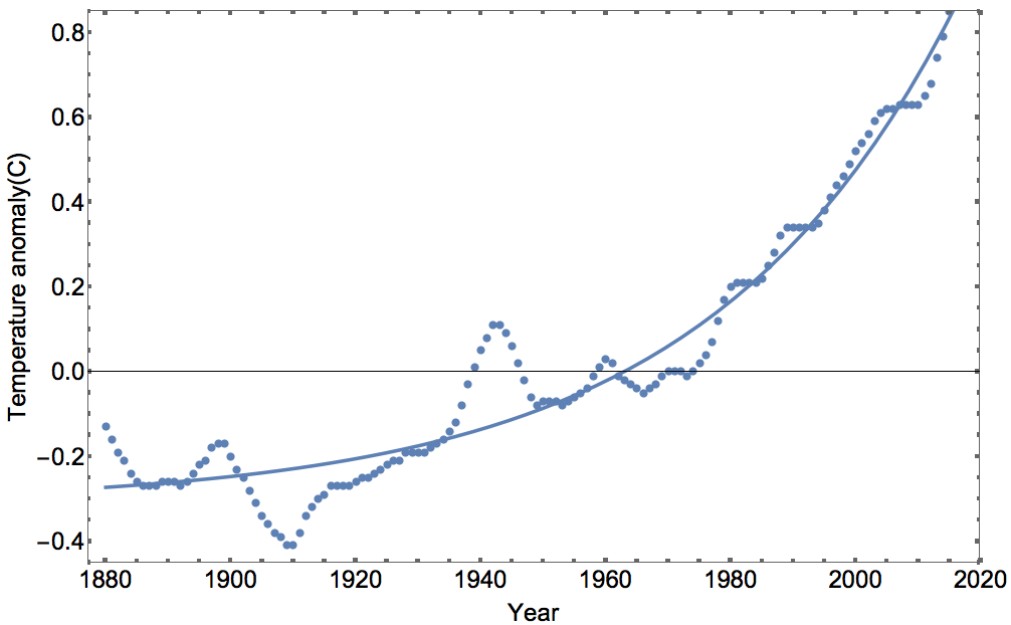

**Figure 2.** Temperature anomaly, the change in the global surface temperature relative to 1951–1980 average temperature (NASA, 2017). Dots represent five-year moving averages; the solid curve is given by Eq. (13).

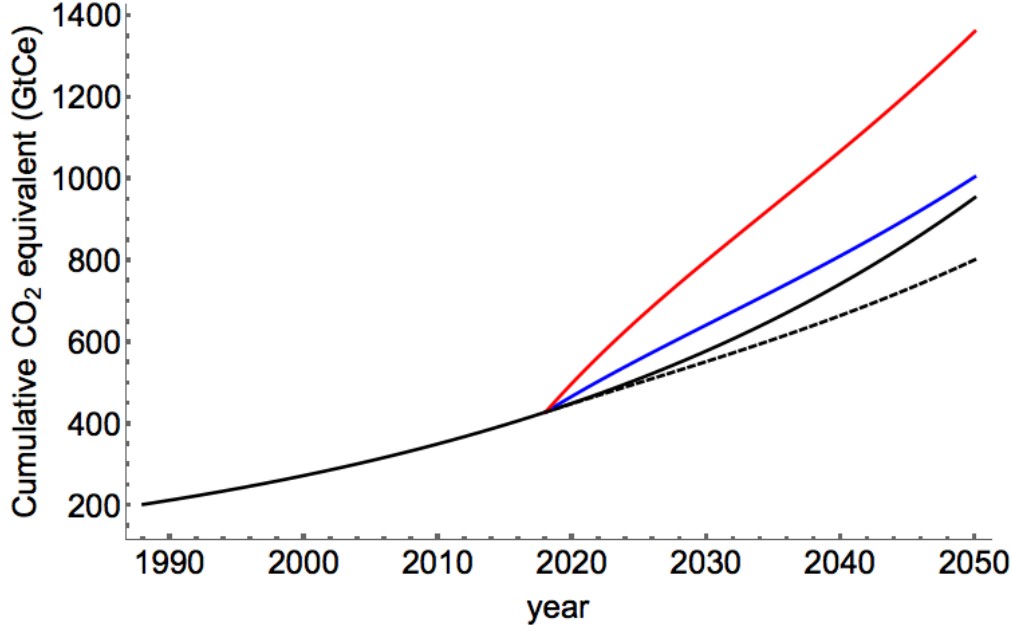

**Figure 3.** Four emission scenarios: (1) Business as usual using coal (black curve); after 2018: (2) $CH_4$ with 6% fugitive (blue); (3) $CH_4$ with 12% fugitive (red curve)); (4) $CH_4$ with critical fugitive fraction, (dashed) 2.2%, as shown in Tab. 1.



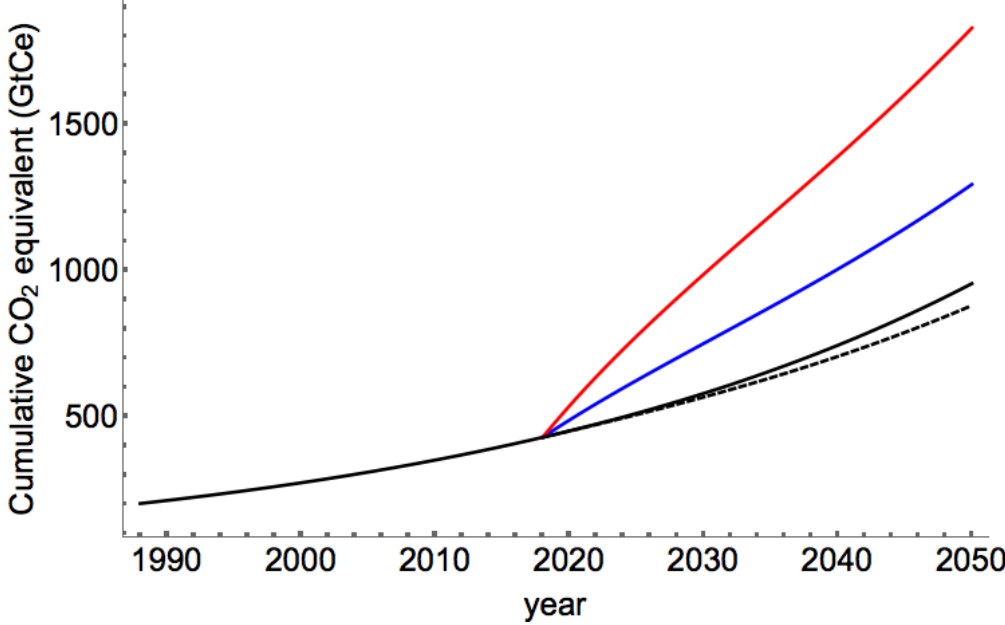

**Figure 4.** Four emission scenarios: (1) Black: business as usual using oil (black curve); after 2018: (2) $CH_4$ with 6% fugitive (blue); (3) $CH_4$ with 12% fugitive (red); (4) $CH_4$ with critical fugitive fraction, (dashed) 0.76%, as shown in Tab. 1.

energy conservation would require global collaboration and redistribution of wealth on unprecedented scale. Fig. 5 shows two pathways that phase out fossil fuels starting in 2018 and are consistent with the proposed Hansen *et al.* budget. The area under both curves, the total $CO_2$ put into the atmosphere is 525 GtC, a number chosen because it happens to reproduce the rates of emission reduction contained in the Hansen *et al.* 2013 paper, *i.e.* 3.5% in 2003, 6% in 2013, and 15% in 2020.

5     As an important aside, beyond the scope of the this paper, is that Hansen *et al.* have concluded, in view the industrialized world's lack of action, that the climate can only be stabilized by "negative emissions," *i.e.,* by extraction of $CO_2$ from the air (Hansen et al., 2017). This approach will be very expensive; it is also fraught with danger because of the difficulty of accurately predicting the instabilities associated with our human time scale irreversible disintegration of ice sheets and ice shelves Hansen (2005).

10    In Fig. 6, the black curve shows cumulative emissions corresponding to a phase-out of fossil fuels following the exponentially decaying pathway, the blue curve in in Fig. 5. The blue curve corresponds to a complete switch-over from coal to $CH_4$ in 2018 with 6% fugitive $CH_4$; the red curve is the analog with 12% fugitive $CH_4$. Fig. 7 differs only in that the switch-over is from oil to $CH_4$. Once again, because the increase in efficiency (4/3) is less in this case, the relative importance of fugitive $CH_4$ is enhanced. In both cases the dashed curves correspond to the respective fugitive fractions of coal and oil.



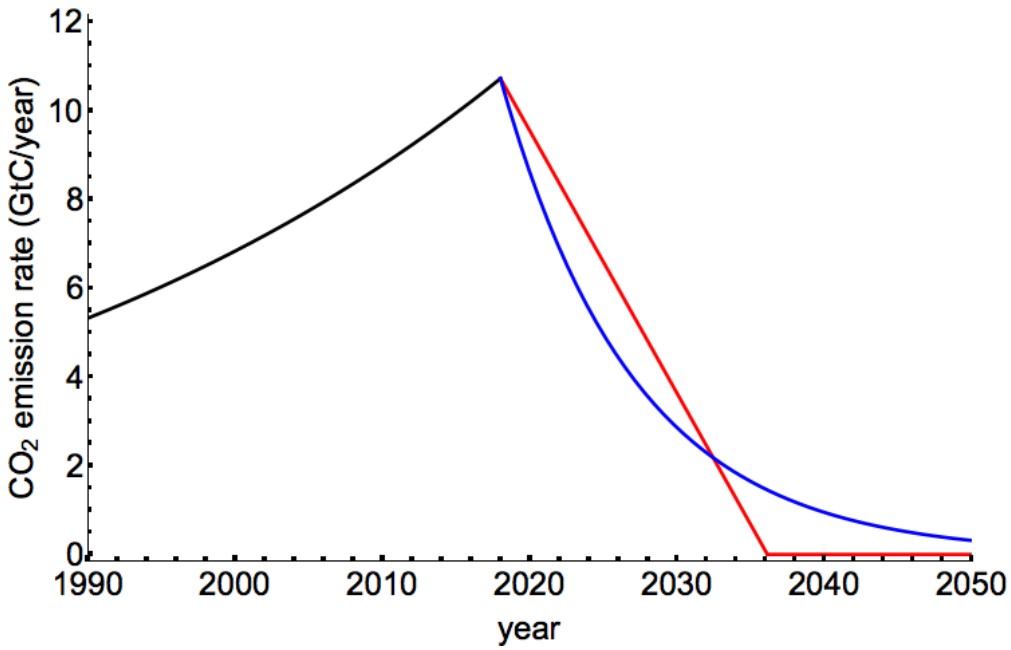

**Figure 5.** Global phase-out of fossil fuels: business as usual until 2018 followed by exponential (blue) and linear decay (red).

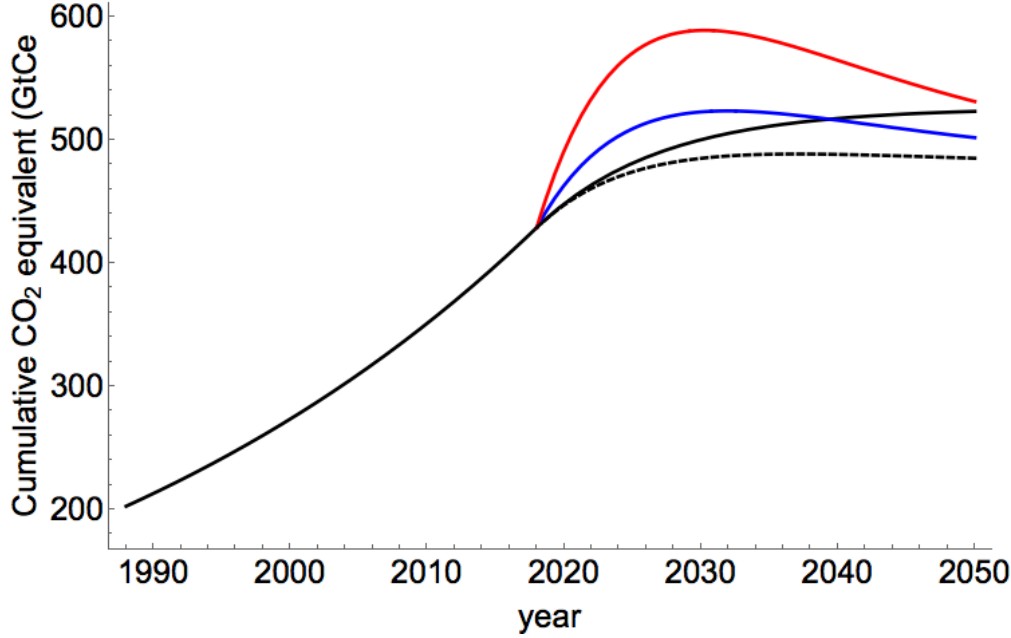

**Figure 6.** Four emission scenarios: exponential phase out of fossil fuel assuming (1) coal (black curve); (2) after 2018: $CH_4$ with 6% fugitive (blue); (3) $CH_4$ with 12% fugitive (red); (4) $CH_4$ with critical fugitive fraction, 2.2% (dashed), as shown in Tab. 1.



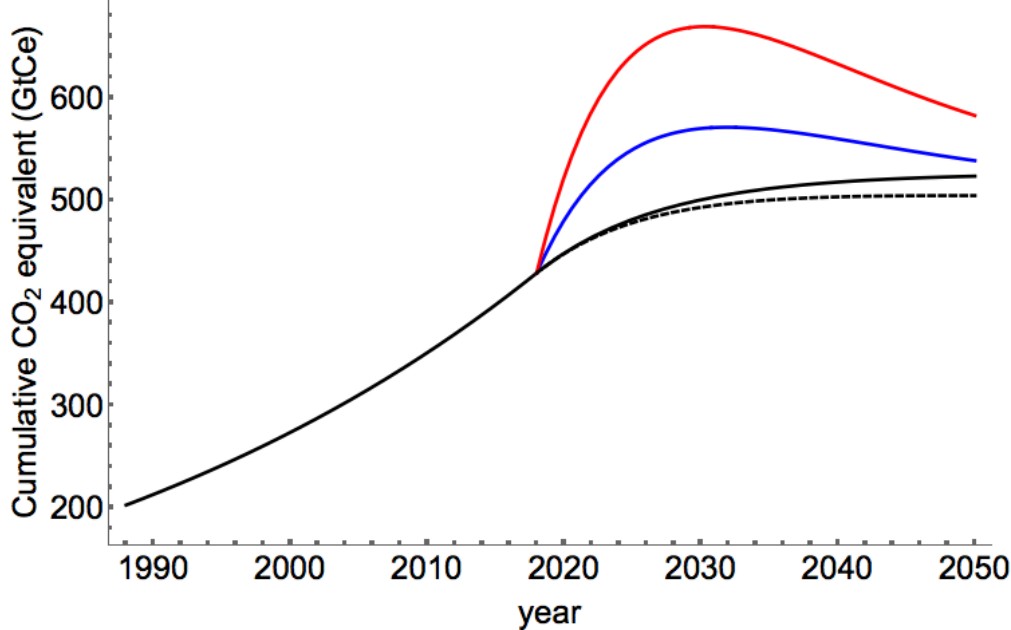

**Figure 7.** Four emission scenarios: Exponential phase out of fossil fuel assuming (1) oil (black curve); after 2018: (2) $CH_4$ with 6% fugitive (blue); (3) $CH_4$ with 12% fugitive (red); (4) $CH_4$ with critical fugitive fraction, 0.76% (dashed), as shown in Tab. 1.

## 4 Conclusions

Figs. 3, 4, 6, and 7 clearly support what has been clear for some time, namely that *"By The Time Natural Gas Has A Net Climate Benefit You'll Likely Be Dead And The Climate Ruined,"* as Joe Romm summarized it in the title of one of his posts (Romm, 2014).

In other words, there is no justification for using the 100-year horizon in energy policy choices involving natural gas. Reporting based on $CO_2$ equivalents using the 100-year horizon, which is standard practice (World Resoures Institute & World Business Council For Sustainable Development), is misleading and irreconcilable with the observed time-scale of developments of the climate system (National Oceanic Atmospheric Administration, a, b; Hansen et al., 2017). As the numerical thought experiments presented in this paper demonstrate, using more a realistic, dynamical approximation—one that is consistent with

the climate time scale—is technically trivial and vital for responsible public policy.

There is general agreement that humanity has a finite carbon budget overspending of which is likely to cause irreparable harm to life on Earth. The Intergovernmental Panel on Climate Change (IPCC) in its Fifth Assessment Report quoted as its estimate for this budget $2900 \, \text{GtCO}_2$ (Intergovernmental Panel on Climate Change, 2014). Accounting for the molar mass ratio ($\frac{12}{44}$) of carbon to carbon-dioxide this corresponds to $800 \, \text{GtC}$. This number rests on the ill-founded, by now mostly

abandoned, assumption that a $2\,^\circ\text{C}$ global mean temperature increase is a "guardrail" that protects life on Earth from the essentially irreversible harm of run-away climate change (Geden, 2015; Friedman, 2015; Knutti et al., 2016). Indeed, over the





last couple of years, it has become increasingly clear that relying on this upper limit violates the precautionary approach of Principle 15 of the 1992 Rio Declaration, a treaty signed and ratified by many countries, including the United States.

Fig. 5 is consistent with $1\,°C$ as the "guardrail," a choice based on paleoclimate and other arguments presented in detail by Hansen *et al.* in (Hansen et al., 2008, 2013; Hansen and Kharecha, 2013; Hansen et al., 2017). This paper presented a simple model to keep track of how much of the global greenhouse gas budget is spent in carbon-equivalent units, a choice that *is* consistent with the precautionary principle.

As Figs. 6 and 7 make clear, there is no rational argument for phasing out fossil fuels while at the same time engaging in a complete replacement of coal and oil power plants by natural gas fired ones. The same, but to an even higher degree, applies to the introduction of natural gas vehicles (See *e.g.* State of Rhode Island Office of Energy Resources, 2015). Nonetheless, that exactly seems to have been the national energy policy of the Obama administration and still is policy of the seemingly more enlightened states in the United States.

*Acknowledgements.* The author is greatly indebted to Professor Randy Watts for his careful reading of an early draft of this paper and for his invaluable suggestions.

*Competing interests.* No competing interests are present.



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
