# Peer review of "Only the instantaneous global warming potential is consistent with honest and responsible greenhouse gas accounting"

_Earth System Dynamics, 2018_

## Referee Comment (RC1) · K. Caldeira (Referee) · 22 May 2018

The title of this paper suggests that its primary function is not to act as a scientific paper but rather to support a normative claim.

Concepts like what it means to be 'honest and responsible' in this context do not lend themselves to empirical tests, but rather express normative judgments.

GWPs are flawed metrics for almost every purpose, so I do not seek here to defend the use of GWPs.

GWPs are metrics, and thus do not have a truth value. Like all tools, they can be more

useful or less useful, but they cannot be right or wrong.

One could perhaps rephrase the title as: "Among global warming potential definitions, only the instantaneous global warming potential is useful", but this too is a matter of degree and depends on what you want to know.

If the author could come up with a cogent argument for what would be better than GWPs as conventionally defined that have not already been discussed in the literature, I would be open to reviewing that as a perspective or opinion piece.

Addressing the normative claims, instantaneous radiative forcing values would seem to be a flawed metric for greenhouse gas accounting and attribution. Imagine two gases with the same instantaneous radiative forcing, but one decays in a year and the other remains in the atmosphere forever. Would it be wise to consider these two gases to be equivalent? The paper criticizes GWPs as conventionally defined but does not make a strong case for the use of instantaneous radiative forcing as an improvement. Indeed, many have criticized GWPs as conventionally defined for not considering effects on long time scales.

The author may want to resubmit as an opinion or perspective piece, but on a quick perusal I would not be enthusiastic to review that.

Ideally, in a policy context, one would like a metric to compare different greenhouse gases that would indicate the relative amount of damage that would be caused by an equal mass release of the different gases. This measure would be is the ratio of the value of the damage caused by release of gas X to the damage caused by release of an equivalent quantity of CO2, where that time series of damage is appropriately turned into a scalar value to allow simple comparison.

Unfortunately, the estimation of future marginal damage and the conversion of time series to scalars (typically done in a net present value calculation) are rife with problems that have been widely discussed already. Further, the relative damage would also

depend on the assumed background scenario against which these emissions occur.

GWPs are used mostly because of historical legacy. They are clearly flawed metrics. Some people use them and are unaware with their deficiencies. Others use them, aware of their deficiencies. There are no doubt dishonest and irresponsible people who use these metrics to try to achieve nefarious ends. But users of flawed GWP metrics can be both honest and responsible.

And scientific papers should report previously unknown empirical facts, not value judgments.

---

## Referee Comment (RC2) · K. Caldeira (Referee) · 22 May 2018

One way of framing this paper as a scientific paper would be to support the claim: "Different emissions scenarios that are equivalent on conventional GWP metrics produce very different climate outcomes."

I am not sure how many papers, if any, already make that point compellingly.

A more useful paper would be to provide a new metric such that different emission scenarios that are equivalent on this new metric would all have very similar climate outcomes.

[Figure]

The focus on instantaneous effects suggested in the title of the Nightengale's work would not satisfy this objective.

---

## Author Comment (AC1) · 30 May 2018

The referee does not seem to object to the science in the paper nor to the claimed relevance of the precautionary principle in the abstract. His main issue seems to be the use of "honest and responsible" in the tittle. In response, I note the following. Line 12 of page 2 of the paper refers to this quote from IPCC's AR5:[1]

> The choice of time horizon has a strong effect on the GWP values—and thus also on the calculated contributions of $CO_2$ emissions by component, sector or nation. There is no scientific argument for selecting 100 years compared with other choices ....The choice of time horizon is a **value judgement** [emphasis added] because it depends on the relative weight assigned to effects at different times.

The same AR5 document also makes it clear that:[2]

> The choice of metric and time horizon depends on the particular application and which aspects of climate change are considered relevant in a given context Metrics do not define policies or goals but facilitate evaluation and implementation of multi-component policies to meet particular goals. **All choices of metric contain implicit value-related judgements such as type of effect considered and weighting of effects over time** [emphasis added].

Ocko *et al.* in a paper in Science sum up the conundrum facing the scientific community as it confronts the widespread, misguided use of scientific tools, GWPs in particular:[3]

> Policy-makers often treat a GWP as a value-neutral measure, but the time-scale choice is central to achieving specific objectives ...."

Indeed, the UNFCCC, and as a consequence the United States Environmental Protection Agency rely on use of the 100-year GWP, as does the United States Department of State, *e.g.*, in its 2014 Climate Action Report.[4] It is very likely that this use will add to "young people's burden: requirement of negative $CO_2$ emissions."[5] I mention this paper because Earth System Dynamics published it while its title, at least to my ears, has an unmistakable, normative ring.

As to the science, the referee writes:

Imagine two gases with the same instantaneous radiative forcing, but one decays in a year and the other remains in the atmosphere forever. Would it be wise to consider these two gases to be equivalent?

Of course not. On the contrary, the model that the paper uses in its thought experiment features two gasses, $CO_2$ and $CH_4$. The former is treated as stable; the second as decaying with a time constant $\tau$, that appears in the equations used to produce Figs. 3, 4, 6, and 7. This is not the best conceivable model, nor is it presented as such, but it does much better job than using the $GWP_{100}$. The model shows (semi-)quantitatively that use of methane as a bridge fuel is irreconcilable with the alarming time scale of current developments of the cryosphere, a dominant time scale of the climate system.

One of the section headings of a paper in The Lancet reads: [6]

**Climate change effects on health will exacerbate inequities between rich and poor**

The use of the word "exacerbate" is clearly normative. Does the referee object to this? Does this use make the statement less empirically testable?

To sum up, both with respect to the relevant time and length scales, the use of the $GWP_{100}$ by the organizations mentioned above lacks scientific justification nor can it be reconciled with the precautionary principle. Indeed, as the referee mentions, "GWPs are flawed metrics for almost every purpose." The paper provides scientific arguments why in these instances use of this metric is misguided; see, for example, the paragraph starting on line 6 of page 3. The referee's objections seem mostly about adjectives—such as the use of "honest" instead of "useful"—rather than about scientific substance.

However that may be, certainly when a livable climate is at stake, it seems reasonable to have as one of the missions of Earth System Dynamics that it focus attention on misguided and irresponsible use of scientific tools, certainly when human behavior in the Anthropocene is an integral part of those dynamics.
* * *
[1] G. Myhre, D. Shindell, F.-M. Bréon, W. Collins, J. Fuglestvedt, J. Huang, D. Koch, J.-F. Lamarque, D. Lee, B. Mendoza, T. Nakajima, A. Robock, G. Stephens, T. Takemura, and

H. Zhang, "Anthropogenic and natural radiative forcing," in *Climate Change 2013: The Physical Science Basis. Contribution of Working Group I to the Fifth Assessment Report of the Intergovernmental Panel on Climate Change*, edited by T. F. Stocker, D. Qin, G.-K. Plattner, M. Tignor, S. K. Allen, J. Boschung, A. Nauels, Y. Xia, V. Bex, and P. M. Midgley (Cambridge University Press, Cambridge, United Kingdom and New York, NY, USA, 2013) Book section 8, pp. 659–740page 711

[2] See page 663 of Ref. 1.

[3] I. B. Ocko, Steven P. Hamburg, Daniel J. Jacob, David W. Keith, Nathaniel O. Keohane, Michael Oppenheimer, Joseph D. Roy-Mayhew, Daniel P. Schrag, and Stephen W. Pacala, "Unmask temporal trade-offs in climate policy debates," Science **356**, 492–493 (2017).

[4] See for example the emission numbers contained in Table 3 on page 18 of [7].

[5] J. Hansen, M. Sato, P. Kharecha, K. von Schuckmann, D. J. Beerling, J. Cao, S. Marcott, V. Masson-Delmotte, M. J. Prather, E. J. Rohling, J. Shakun, P. Smith, A. Lacis, G. Russell, and R. Ruedy, "Young people's burden: requirement of negative $CO_2$ emissions," Earth System Dynamics **8**, 577–616 (2017).

[6] See page 1694 of "Managing the health effects of climate change," Lancet **373**, 1693–1733 (2009).

[7] "United States Climate Action Report 2014," (2014).

---

## Author Comment (AC2) · 31 May 2018

The paper is clearly intended to correct a widespread, misguided, and irresponsible use of the GWP. Is the previous statement a value judgement? Yes, it is but it also is shorthand for an experimentally verifiable statement. Just rephrase it as a conditional statement assuming a purpose shared by most of humanity.

Scientific journals and their editorial policy determine the direction of scientific development. Science ideally is objective; the direction in which it develops is not: a scientific journal with jellyfish as editors would put a very different spin on ocean acidification.

[Figure]

"Instantaneous" in the title of the paper is appropriate. The paper discusses the GWP, and shows that only the instantaneous version can be used to produce useful greenhouse gas inventories in terms of carbon equivalents. I discussed this in detail in my previous referee comment. Nevertheless, that is what this tool is commonly used for. The experimental test of how detrimental that is for life on earth is being performed as we speak.

If one's purpose is to make the most accurate predictions, one should use the best possible models. However, if the purpose is to change misguided policies based on failed heuristic metrics, it seems to me to be better to use simple arguments. Although the mathematics may still be beyond the grasp of most policy makers, this is what the paper tries to do. In the process, as mentioned before, the paper shows that there is a serious time scale issue having to do with the virtually impossible to quantify crossing of climate tipping points.

All of the above is well-known, but obviously human behavior, an important determinant of climate dynamics, is not informed by those who know this well. It seems to me that it is the role of scientific journals to play a peer-reviewed role in this correcting this.

---

## Referee Comment (RC3) · Anonymous Referee #2 · 20 Jun 2018

I recommend that this paper is rejected.

The study is well motivated but flawed. I had expected (from the abstract) to find some coherent reason why the instantaneous GWP is superior to the normal GWP(100). However, all I find (p 3;l 18-19) is an assertion that this IS the case and then the rest of the paper follows as if that assertion is justified. In fact the abstract contains no useful information about the content of paper, but only really states the assertion.

I am no great fan of the GWP and the difficulties of using it to represent temperature change have long been known (its equivalence is formally restricted to time-integrated radiative forcing following a pulse emission). See for example Figure 3 of Fuglestvedt

et al. (Climatic Change 58, 267-331, 2003) and many of the figures and references in Myhre et al. (2013).

There is much I disagree with in this paper, but I restrict myself to those aspects that I feel justify the rejection.

The principal problem is that no account is taken of the much greater persistence time of CO2 perturbations, especially the fact that some of that CO2 is an essentially permanent addition to the atmosphere. This is acknowledged at p 2;l 28-29, but plays no subsequent part in the analysis. The only timescale used in the paper is methane's decay time.

The problem with the key figures (Figs 3 and 4) is that they just demonstrate the result of applying the assertion, rather than demonstrating that the assertion leads to a better representation of the resulting climate change than applying GWP(100), which is surely what matters. If the temperature effects (a simple physical model could be used in an illustrative context)) of using CO2-equivalents calculated using the GWP(0) was adopted, and compared with that resulting from the actual emissions (in the author's thought experiment) the temperature evolution of actual and CO2-equivalent emissions would be quite different. The impact of methane emissions from any given year would decay to near zero in a few decades, while much of its (large) equivalent in terms of CO2 using GWP(0) would remain in the atmosphere influencing climate for long periods.

The author invokes the precautionary principle but this only applies if the chosen metrics have demonstrable integrity. By placing a very large multiplier on CH4 emissions, it would encourage large cuts to methane emissions in preference to those of CO2, but the longer-term consequences of such a choice would have to be explored to assess the extent to which such a policy is precautionary or ultimately leads to a greater climate change (which could only be reversed by the negative emissions that the author (p 9;l 7) regards as "fraught with danger").

The discussion surrounding Figures 1 and 2 is confused – again we are left with an assertion that the similarity between the Figures show consistency, when such consistency can only be demonstrated by converting emissions to changes in concentrations, changes in concentrations to radiative forcings and (transient) radiative forcing to (transient) changes in temperature. To do otherwise is to ignore the physics of the climate system. In essence, the attribution statements in IPCC AR5 are tracing through those necessary links.

---

## Author Comment (AC3) · 25 Jun 2018

The referee writes:

> The principal problem is that no account is taken of the much greater persistence time of $CO_2$ perturbations, especially the fact that some of that $CO_2$ is an essentially permanent addition to the atmosphere. This is acknowledged at p 2;l 28-29, but plays no subsequent part in the analysis. The only timescale used in the paper is methane's decay time.o

The thought experiment described in this paper features two greenhouse gases: $CO_2$ and $CH_4$. Both are described mathematically by Eq. (3), which has a homogeneous decay term and an inhomogeneous source term. $CH_4$ has a finite lifetime $\tau = 12.4$ year; the lifetime of $CO_2$ is approximated by $\infty$, exactly as the referee expects.

This infinite lifetime appears in the denominator and is consequently mis-identified by the referee as missing from the analysis. Indeed, the equation that ultimately determines the time-development of the carbon-equivalent concentration $c_e$ lacks a decay term. It only features a source term due to the combustion of $CH_4$ and the carbon-equivalent effect of the fugitive $CH_4$. Both the infinite time scale of $CO_2$ and the finite one of $CH_4$ are not only present in the analysis; they are crucial. In other words, the statement the referee makes in the last sentence of the comment quoted above is incorrect.

One might criticize the paper for not using better Green functions for $CO_2$ and $CH_4$, but in the thought experiment I chose for simplicity for clarity's sake, rather than for accuracy. This seems to have confused the referee. Nevertheless, crude as it may be, the model used in the paper is a vast improvement over the widespread, misleading use of the 100-year time horizon used by the UFCCC and the U.S. Environmental Protection Agency.

Let me reiterate that the main results of the thought experiment described in this paper are as follows:

1. The paper provides a rough estimate of how long it will take to a see reduction of global warming due to switching to $CH_4$ with its higher energy content relative to coal and oil. This cross-over shows up in the carbon-equivalent greenhouse gas concentration illustrated in Figs. 3, 4, 6, and 7. In the long run, the red and blue $CH_4$ curves all intersect the black coal/oil curves. Except in one case, as shown in Fig. 6, these cross-over points are too far into the future to be relevant for decision makers and do they not actually show up in the figures.

2. The elementary kinetic equations used in the paper show that any global warming potential with a greater than zero time horizon precludes a time-dependent analysis as presented. As a consequence, not even a rough estimate of the cross-over time can be made on the basis of these quantities.

Since the "principal problem" the referee claims to have identified a problem does not exist, it seems pointless to address secondary issues brought in this report.

---

## Author Comment (AC4) · 20 Aug 2018

**Proposed policymaker-friendly metric of radiative effects of greenhouse gases**

Peter Nightingale

Department of Physics, University of Rhode Island, Kingston, Rhode Island 02881, U.S.A.

**Correspondence:** Peter Nightingale (nightingale@uri.edu)

**Abstract.**

This paper proposes a simple metric for the dynamic evaluation of the cumulative, combined impact on global warming of greenhouse gasses. As an illustration, the metric is applied to methane (natural gas) when used for energy production. The proposed metric accounts for the effect on a decadal timescale of energy policies based on natural gas as a purported bridge fuel.

5 Results of a thought experiment evaluated by the proposed metric explicitly show problematic policy aspects of the commonly employed global warming potential of methane with a 100-year time horizon which:

**1:** lacks a solid scientific basis and is incompatible with crucial timescales;

**2:** does not allow for continuous-time dynamic tracking of greenhouse gas emissions; and

**3:** is incompatible with the Precautionary Principle.

10 ## 1 Introduction

Hansen et al. (2008) argue that atmospheric $CO_2$ concentration exceeding 350 ppm poses a serious threat to human existence and life on earth in general. As the authors put it in their abstract:

> If humanity wishes to preserve a planet similar to that on which civilization developed and to which life on Earth
> is adapted, paleoclimate evidence and ongoing climate change suggest that $CO_2$ will need to be reduced from its
15 > current 385 ppm to at most 350 ppm, but likely less than that. The largest uncertainty in the target arises from
> possible changes of non-$CO_2$ climate forcings.

The paper warned that continued growth of greenhouse gas emissions for just another decade—after 2008—would make it practically impossible to avoid catastrophic effects on the climate system. A decade later, atmospheric $CO_2$ is fluctuating around 410 ppm (Pro Oxygen, 2018) and it still appears to be increasing at a rate roughly in the range of 2–2.5 percent per 20 year.

In the aforementioned quote, the authors qualify the critical number, 350 ppm, mentioning that it is likely too high, as it fails to account fully for the dangers of non-$CO_2$ forcings. Of these, atmospheric methane is dominant. What is more, methane forcing today is far more impactful than it was in 2008. For instance, Turner et al. (2016) concluded on the basis of satellite

and surface data that there had been a large increase in the methane emissions of the United States over the decade prior to their study. Worden et al. (2017) subsequently traced this increase back to fossil fuel sources.

At the same time, as numerous publications over the last decade have made clear, the climate system is changing at a rate outpacing projections, those of the Intergovernmental Panel on Climate Change (IPCC) in particular. For instance, Rahmstorf et al. (2007) mention that projections may have underestimated changes in sea level rise. Hansen et al. (2013) mention that end of summer Arctic sea ice has been declining a factor of four faster than in IPCC models. Also the *Third National Climate Assessment* (Melillo et al., 2014) states that the "only real surprises have been that some changes, such as sea level rise and Arctic sea ice decline, have outpaced earlier projections." Brown and Caldeira (2017) discuss rapid nonlinear melting of the Greenland and Antarctic ice sheets not represented in IPCC model assessments.

There are numerous other such under-predicted developments such as, for example, the Arctic amplification documented in recent *Arctic Report Cards* issued by the National Oceanic Atmospheric Administration (a, b). Underestimates should not come as a surprise. Indeed, Brysse et al. (2013) *et al.* discuss a series of examples of scientists "erring on the side of least drama."

Developments of the cryosphere clearly have a large decadal component and indeed, as Steffen et al. (2018) and also Rintoul et al. (2018) have argued, decisions made during the next one or two decades may lead to irreversible changes of the climate system. Nonetheless, and in spite of critical observations of IPCC going back to its *Second Assessment Report* (Houghton et al., 1995), the global warming potential (GWP) with a 100-year time horizon has become the metric employed— pursuant to the United Nations Framework Convention on Climate Change (UNFCCC)— to assess public policy with respect to multi-gas (usually called $CO_2$ equivalent) emissions.

With the considerations in mind it should be noted that IPCC's *Fifth Assessment Report* (AR5) explicitly states that there is no scientific argument for using the 100-year GWP horizon—see, *e.g.,* page 711 of Stocker et al. (2013).[1] In fact, as AR5 puts it: "All choices of metric contain implicit value-related judgements such as type of effect considered and weighting of effects over time." Note against this background that Ocko et al. (2017) have pressed for more transparency in climate policy issues with respect to the often hidden implied temporal trade-offs.

The climate system of the earth is a complex system far from thermodynamic equilibrium with many inseparable time- and lengthscales. In such a system, uncontrolled, scientifically hard to justify approximations will always characterize any attempt to isolate simple metrics for use by policy makers to gauge—as was IPCC's design purpose—the relative radiative effects of divers greenhouse gasses.

More specifically, as argued above, the disruption of the climate system of the earth and the human role in it clearly have important decadal timescale features. Given this, use of the 100-year horizon as the basis of major energy policy decisions has no basis in science. Whatever value judgments may have led to general acceptance of this 100-year metric, it appears to be irreconcilable with the Precautionary Principle, number 15 of the Rio Declaration of the United Nations General Assembly; United Nations Change Change. For further discussion of this see Section 2.
* * *
[1]The following is a representative list of comments about the global warming potential to be found in various IPCC assessments: Houghton et al. (1995) pages 21 and 73; and Stocker et al. (2013) pages 58, 663, 710, and 711.

In addition to these general considerations and because of the shift in the United States over the last decade to natural gas,[2] which is used increasingly for the generation of electricity, it is imperative to provide policymakers with tools that do not downplay the effects of non-$CO_2$ emissions with very strong near-term effects on the climate.

The simple dynamical metric proposed here produces order of magnitude estimates based the instantaneous global warming potential rather than the one based on the 100-year horizon. The results show that simple, user-friendly alternatives exist for the 100-year time horizon global warming potential. Edwards and Trancik (2014) presented a similar approach, one that also focuses on a dynamical approach rather than the static one implicit in the use of any non-instantaneous global warming potential.

The layout of the paper is as follows: Section 2 reviews some of the well-known basic properties of methane and introduces the proposed simple dynamical metric. Section 3 presents results of some simple energy policy thought experiments. Finally, Section 4 summarizes the conclusions.

**2   Methane basics**

The global warming potential, as mentioned in Section 1, is a simple tool designed to estimate the relative effect of greenhouse gasses on global warming. It was designed and accepted to assist in policy making. This quantity is a dimensionless multiplier that converts the effect of the emission of a unit mass pulse of a greenhouse gas under consideration to a mass of $CO_2$ that would have the same global warming effect, the $CO_2$ equivalent ($CO_2$e) mass. A pulse of $CO_2$ injected into the atmosphere is taken up by the ocean, biosphere and soil and decays by half in about 25 years but 20% is still in the atmosphere after 500 years; see Fig. 4A in Hansen et al. (2013). Atmospheric $CH_4$, on the other hand, has a half-life of less than a decade.

More explicitly, the global warming potential, as defined in Section 8.7.1.2 of Stocker et al. (2013), is a fraction: the time-integral of the radiative forcing due to a pulse emission of a given greenhouse gas divided by same quantity for a pulse of an equal mass of $CO_2$. Due to the atmospheric dynamics of both $CO_2$ and $CH_4$ the resulting global warming potential depends on the time interval used in the integrals, aka the time horizon. The global warming potential is denoted by $G_t$ with $t$ the time horizon in units of years.

The choice of the time horizon $t$ is a major source of arbitrariness. In addition to the value judgment mentioned in Section 1 and acknowledged by the Intergovernmental Panel on Climate Change (IPCC), there is the issue of the timescales relevant to the physical process and policy decisions under consideration.

For a project small on a global scale, averaging emissions over the expected life time of that project might make physical sense, but for matters of global scale, such as the energy policy major global greenhouse gas emitting nations, the horizon should be set by the timescale of the global climate change phenomena and the danger they pose to life on earth. Therefore, as mentioned in Section 1, there notably are the following considerations, among others:
* * *
[2]Exploration and production subsidies from the federal government have increased dramatically, during the Obama administration; see, *e.g,* Oilchange International. This trend is expected to accelerate during the Trump administration.

1. The arguments made by Hansen (2005) and the well-known difficulty of predicting instabilities (aka state shifts or tipping points) such as the sudden and, on a human multi-generational timescale irreversible, disintegration of ice sheets; *i.e.,* as Drijfhout et al. (2015) put it, the fact that tipping points "notoriously difficult to foresee;"

2. Recent developments on a decadal timescale in the Arctic (National Oceanic Atmospheric Administration, a, b);

3. The fact that decisions made in the next one or two decades may determine the fate of the future of Antarctica and the Southern Ocean, as argued by Rintoul et al. (2018) argue, or set the climate system on a for all practical purposes irreversible trajectory to what Steffen et al. (2018) refer to as "Hothouse Earth;"

4. The international treaty obligation of the Precautionary Principle 15 of Rio Declaration mentioned in the Section 1 (United Nations General Assembly; United Nations Change Change).

10    Based on these matters, and the simple, mathematical fact that non-instantaneous global warming potentials cannot be used straightforwardly in a dynamical approach, the metric proposed in this paper uses the instantaneous global warming potential $G_0$, the instantaneous radiative forcing relative to that of $CO_2$.

[revised manuscript text omitted]

An additional assumption made in the choice of this metric is that $CO_2$ is treated as an atmospheric gas with an an *infinite decay time.* In other words, for $CO_2$ the first term on the right-hand side of Eq. (3) vanishes, so that $CO_2$ evolves by simply adding up for ever. The justification for this approximation is that, as shown by Matthews et al. (2009), the total allowable emissions, *i.e.,* the budget for climate stabilization, is approximately independent of the time and place of those emissions. At the same time, the metric developed here is set up so that policy makers can track the expenditures to be charged to that budget as a result of their policies.

The final result for the atmospheric $CO_2$ equivalent concentration at time $t > 0$ is obtained by substituting Eqs. (8) and (9) into Eq. (5). Subject to the specified initial conditions, the solution of the differential equations is:

$$c_e(t) = \int_{-\infty}^0 p(t)\, dt + \frac{1}{\varepsilon}\int_0^t p(t_1)\, dt_1 + \frac{f}{(1-f)\varepsilon}\left[ G_0' \int_0^t e^{\frac{t_1-t}{\tau}} p(t_1)\, dt_1 + \frac{1}{\tau}\int_0^t \int_0^{t_1} e^{\frac{t_2-t_1}{\tau}} p(t_2)\, dt_2\, dt_1 \right].$$ (10)

Note that the first two terms represent the cumulative emissions since the Industrial Revolution, approximated here as having occurred at $t = -\infty$ and the additionally accumulated amount as of $t = 0$, when the in this thought experiment hypothetical switch to $CH_4$ occurs. The third term represents the $CO_2$-equivalent of the accumulated fugitive $CH_4$. The fourth term accounts for the accumulated $CO_2$ by oxidation of fugitive $CH_4$, oxidized at various times starting at $t = 0$.

In the case discussed in this paper, the function $p$ is represented accurately by a simple exponential, as shown in the next Section 3, so that the integrals can be done exactly; in more complicated cases, numerical integration is straightforward. In practical applications of a dynamical scheme of this sort, it would suffice to use a finite-difference approximation based on yearly data and appropriately chosen initial conditions.

**3  Results**

Estimates of total carbon dioxide emissions from the beginning of the Industrial Revolution are available from the Carbon Dioxide Information Analysis Center (CDIAC) (2014). As shown in Fig. 1, the data can be represented surprisingly accurately by a simple exponential growth curve; the curve shown in Fig. 1 satisfies the equation

$$C_{\text{global}}(t) = 9.00\, e^{0.025\left(\frac{t}{\text{year}} - 2010\right)}\ \text{GtC/year}.$$ (11)

[Figure]

**Figure 1.** Global $CO_2$ emissions in gigatons of carbon per year with exponential fit, Eq. (11).

This equation is used to define *business-as-usual*. The expression was obtained by a least squares fit, followed by a slight adjustment of the normalization constant so that the integral from $-\infty$ to year 2011 reproduces

$$\int\limits_{-\infty}^{2011\,\text{year}} C_{\text{global}}(t)\,dt = 365\,\text{GtC}, \tag{12}$$

the CDIAC estimate of 2011 cumulative emissions.

5   Given that $CO_2$ emissions are the predominant driver of global warming, it is not surprising that temperature anomaly $T$, shown in Fig. 2, is consistent with the climate forcing resulting from these emissions. The temperature anomaly data of NASA/GISS (NASA, 2017) can be used for a linear regression, two-parameter least-squares fit using the same exponential function featured in Eq. (11). This yields the following expression

$$T(t) = -0.3\,^\circ\text{C} + 1.0\,^\circ\text{C}\,e^{0.025\left(\frac{t}{\text{year}} - 2010\right)}, \tag{13}$$

10   shown as the solid curve in Fig. 2.

Here are the results of one thought experiment: assume, first of all, that business-as-usual continues and that global energy consumption keeps growing exponentially, and, secondly, that power is generated by combustion of coal or oil before 2018 and of $CH_4$ after that, corresponding to time $t = 0$ in Section 2 and the vanishing upper limit in the first integral and lower limits of the integrals in Eq. (10).

15   This produces Fig. 3 in which the solid black curve on the left represents the actual, historical development, a trajectory continued on the right. The blue curve starting in 2018 corresponds to a hypothetical, complete switch to $CH_4$ in that year with 6% of the $CH_4$ escaping unburned, *i.e.,* half of the estimate in Howarth (2015). The red curve corresponds to 12% fugitive $CO_2$. Also included is a black-dashed curve for the critical fraction of fugitive methane as specified in Tab. 1. Fig. 4 is the

[Figure]

**Figure 2.** Temperature anomaly, the change in the global surface temperature relative to 1951–1980 average temperature (NASA, 2017). Dots represent five-year moving averages; the solid curve is given by Eq. (13).

[Figure]

**Figure 3.** Four emission scenarios: (1) Business-as-usual using coal (black curve); after 2018: (2) $CH_4$ with 6% fugitive (blue); (3) $CH_4$ with 12% fugitive (red curve)); (4) $CH_4$ with critical fugitive fraction, (dashed) 2.2%, as shown in Tab. 1.

same assuming that combustion of oil generates power before 2018. Because the efficiency increase is considerably less in this case, the deleterious effect of the fugitive $CH_4$ is more pronounced.

Of course real life is not quite as simple as this thought experiment. However that may be, the results strongly suggest that, although the red and blue $CH_4$ curves will ultimately cross the black coal or oil curves, this does not happens sufficiently
5  rapidly, *i.e.* within one or two decades, to justify the purported role of $CH_4$ as a bridge fuel.

[Figure]

**Figure 4.** Four emission scenarios: (1) Black: business-as-usual using oil (black curve); after 2018: (2) $CH_4$ with 6% fugitive (blue); (3) $CH_4$ with 12% fugitive (red); (4) $CH_4$ with critical fugitive fraction, (dashed) 0.76%, as shown in Tab. 1.

Business-as-usual is one pathway, another one is to stay within a finite carbon budget. Fig. 5 shows two pathways to phase out fossil fuels starting in 2018. These pathways are consistent with the carbon budget proposed by Hansen et al. (2013). The

[Figure]

**Figure 5.** Global phase-out of fossil fuels: business-as-usual until 2018 followed by exponential (blue) and linear decay (red).

area under both curves starting at $t = -\infty$, that is the total $CO_2$ put into the atmosphere, is 525 GtC, a number chosen because it reproduce the rates of emission reduction contained in the Hansen et al. (2013) paper, *i.e.* 3.5% in 2003, 6% in 2013, and 15% in 2020.

In Fig. 6, the black curve shows cumulative emissions corresponding to a phase-out of fossil fuels following the exponentially

[Figure]

**Figure 6.** Four emission scenarios: exponential phase out of fossil fuel assuming (1) coal (black curve); (2) after 2018: $CH_4$ with 6% fugitive (blue); (3) $CH_4$ with 12% fugitive (red); (4) $CH_4$ with critical fugitive fraction, 2.2% (dashed), as shown in Tab. 1.

decaying pathway, the blue curve in in Fig. 5. The blue curve corresponds to a complete switch-over from coal to $CH_4$ in 2018

[Figure]

**Figure 7.** Four emission scenarios: Exponential phase out of fossil fuel assuming (1) oil (black curve); after 2018: (2) $CH_4$ with 6% fugitive (blue); (3) $CH_4$ with 12% fugitive (red); (4) $CH_4$ with critical fugitive fraction, 0.76% (dashed), as shown in Tab. 1.

with 6% fugitive $CH_4$; the red curve is the analog with 12% fugitive $CH_4$. Fig. 7 differs only in that the switch-over is from

oil to $CH_4$. Once again, because the increase in efficiency (4/3) is less in this case, the relative importance of fugitive $CH_4$ is more pronounced. In both cases the dashed curves correspond to the respective fugitive fractions of coal and oil.

**4   Conclusions**

Overspending the carbon budget (mentioned in Section 2) while maintaining a for humans habitable climate is unlikely to be
5   compatible with the time table imposed by the laws of nature. The required replacement of fossil fuels by renewables and energy conservation requires global collaboration and redistribution of wealth on an unprecedented scale. In this context it is worth noting that Hansen et al. (2017) have concluded, in view the industrialized world's lack of action since Hansen et al. (2013), that the climate can only be stabilized by "negative emissions," *i.e.,* by extracting $CO_2$ from the atmosphere.

As illustrated in Figs. 3, 4, 6, and 7, application of the policymaker-friendly tool proposed in this paper—a tool based on
10   the instantaneous global warming potential—clearly supports what has been clear for some time, namely that *"By The Time Natural Gas Has A Net Climate Benefit You'll Likely Be Dead And The Climate Ruined,"* as Joe Romm summarized it in the title of one of his post (Romm, 2014).

In other words, the order-of-magnitude time estimates implied by the graphs presented in Section 3 underscore that there is no scientific justification for using the 100-year horizon in energy policy choices involving natural gas as a bridge fuel. Indeed,
15   reporting based on $CO_2$ equivalents using the 100-year horizon, which is standard practice (World Resoures Institute & World Business Council For Sustainable Development, 2013), obscures short-term effects and is irreconcilable with both the observed timescale of developments of the climate system (National Oceanic Atmospheric Administration, a, b; Hansen et al., 2017) and with that of policy making, a point made by Steffen et al. (2018); Rintoul et al. (2018).

Employing the proposed policy tool, the numerical thought experiments presented in Section 3 demonstrate that using more a
20   realistic, continuous-time dynamic approximation—one that is consistent with the timescale of climate change—is technically straightforward. At the same time, such a tool that respects the relevant timescales may be pivotal in public policy making that stands a chance of preserving a habitable climate for present and future generations.

There is general agreement that humanity has a finite carbon budget overspending of which is likely to cause irreparable harm to life on earth. The Intergovernmental Panel on Climate Change (IPCC) in its Fifth Assessment Report (AR5) quoted as its
25   estimate for this budget $2900\,GtCO_2$ (Intergovernmental Panel on Climate Change, 2014). Accounting for the molar mass ratio ($\frac{12}{44}$) of carbon to carbon-dioxide this corresponds to $800\,GtC$. This number rests on the ill-founded, by now mostly abandoned, assumption that a $2\,°C$ global mean temperature increase is a "guardrail" that protects the biosphere from the essentially irreversible harm of run-away climate change (Geden, 2015; Friedman, 2015; Knutti et al., 2016). Indeed, the climate science research over last decades implies that relying on this upper limit is irreconcilable with the precautionary approach of Principle
30   15 of the 1992 Rio Declaration, a treaty signed and ratified by many countries, including the United States (National Oceanic and Atmospheric Administration—Office of General Counsel).

Fig. 5 is consistent with $1\,°C$ as the "guardrail," a choice based on paleoclimate and other arguments presented in detail by Hansen *et al.* in (Hansen et al., 2008, 2013; Hansen and Kharecha, 2013; Hansen et al., 2017). This simple policy tool presented

here can keep track of how much of the global greenhouse gas budget is spent in carbon-equivalent units, defined in a way that that *is consistent* with the Precautionary Principle.

As Figs. 6 and 7 make clear, there no scientific argument can be made for phasing out fossil fuels while at the same time engaging in replacing coal and oil power plants by natural gas-fired ones. The same, but to an even higher degree—as is clear from the critical fugitive fractions in Tab. 1— applies to the introduction of natural gas vehicles, a conclusion supported by a "pump-to-wheels" study by Clark et al. (2017) that does not take into consideration the full life-cycle, "wells-to-wheels" emissions associated with propulsion.

*Acknowledgements.* The author is greatly indebted to Professor Randy Watts for his careful reading of an early draft of this paper and for his invaluable suggestions. This paper is dedicated to the memory of Robert Malin.

*Competing interests.* No competing interests are present.

---

## Author Comment (AC5) · 20 Aug 2018

**REVISIONS ITEMIZED AND EXPLAINED**

In response to the feedback contained in the reports of the referees and the editor I have revised the presentation of the paper as is evident in the new title "Proposed policymaker-friendly metric of radiative effects of greenhouse gases." More specifically, there are the following changes:

- The paper is presented as an attempt to construct a tool for policy makers. Indeed, the IPCC for the longest time has done exactly that with respect to the global warming potential (GWP). The paper presents arguments to show that the time has come for science to provide a better tool than the UNFCCC-blessed, widely used GWP with a 100-year horizon. In view of comments contained in the referee reports, this should not be controversial in the least.

  The abstract, introduction (Sections 1) and the conclusions (Section 4) have been changed to reflect this. To be more specific, I included new references; see *e.g.* Refs. 1 and 2 and to references to recent, 2018 papers that highlight the decision timeframe: see page 2 paragraph at 15.

- I expanded the discussion about outpaced climate change projections and "erring on the side of least drama" by including more context and additional references, such as Ref 3 and 4. There also is a new reference on tipping points—Ref. 5 in addition to *e.g.* the Hansen Ref. 6, which had already been included.

- The new version of the paper features a more extensive discussion and detailed references to IPCC's own criticism of the GWP going back to it Second Assessment: see page 2 paragraph at 20 and specific page references in footnote 1 on the same page. Also included to provide context is a reference to a paper the title of which is *"Unmask temporal trade-offs in climate policy debates;"* see Ref. 7 in Science. All of this makes it clear that the proposal contained in this paper is part of an ongoing scientific discussion.

- The issue of values and of statements that cannot be objectively confirmed and are not strictly verifiable or falsifiable comes with the territory of trade-offs and policy tools. Words such as "honest" and "responsible" which seem to have come across as inflammatory have been replaced by more neutral ones.

- As to the science background, the current presentation more explicitly addresses the fact that creating a simple tool useful for decision makers is intrinsically problematic. This is traced back to the the wide range of inextricable length- and timescales that characterize a complex system such as the earth's climate in the dynamics of which human behavior and values play a crucial role. For comments in this context about time- and lengthscale inseparability see page 2, paragraph at 25.

- The issue raised by one of the referees, namely that there is only one timescale in the kinetic model, that of methane, has been addressed and clarified in several places. Indeed, the paper features two greenhouse gasses in its calculations: one with an an infinite decay time, namely $CO_@$, and the other on $CH_4$, with a finite decay time. See page 6, paragraph at 10; also see the paragraph at 15 and more specifically the newly added Ref. 8 to justify the treatment of the decay time of $CO_2$ as infinite.

[1] Will Steffen, Johan Rockström, Katherine Richardson, Timothy M. Lenton, Carl Folke, Diana Liverman, Colin P. Summerhayes, Anthony D. Barnosky, Sarah E. Cornell, Michel Crucifix, Jonathan F. Donges, Ingo Fetzer, Steven J. Lade, Marten Scheffer, Ricarda Winkelmann, and Hans Joachim Schellnhuber, "Trajectories of the earth system in the anthropocene," Proceedings of the National Academy of Sciences (2018), 10.1073/pnas.1810141115, http://www.pnas.org/content/early/2018/07/31/1810141115.full.pdf.

[2] S R Rintoul, S L Chown, R M DeConto, M H England, H A Fricker, V Masson-Delmotte, T R Naish, M J Siegert, and J C Xavier, "Choosing the future of antarctica," Nature **558**, 233–241 (2018).

[3] J. M. Melillo, T. C. Richmond, and G. W. Yohe, eds., *Climate Change Impacts in the United States: The Third National Climate Assessment* (U.S. Government Printing Office, 2014) pp. 1–841.

[4] Patrick T. Brown and Ken Caldeira, "Greater future global warming inferred from earth's recent energy budget," Nature **552**, 45–50 (2017).

[5] Sybren Drijfhout, Sebastian Bathiany, Claudie Beaulieu, Victor Brovkin, Martin Claussen, Chris Huntingford, Marten Scheffer, Giovanni Sgubin, and Didier Swingedouw, "Catalogue of abrupt shifts in intergovernmental panel on climate change climate models," Proceedings of the National Academy of Sciences , E5777–E5786 (2015), http://www.pnas.org/content/112/43/E5777.full.pdf?with-ds=yes.

[6] J. Hansen, "A slippery slope: how much global warming constitutes 'dangerous anthropogenic interference'?" Climatic Change **68**, 269–279 (2005).

[7] I. B. Ocko, Steven P. Hamburg, Daniel J. Jacob, David W. Keith, Nathaniel O. Keohane, Michael Oppenheimer, Joseph D. Roy-Mayhew, Daniel P. Schrag, and Stephen W. Pacala, "Unmask temporal trade-offs in climate policy debates," Science **356**, 492–493 (2017).

[8] H. Damon Matthews, Nathan P. Gillett, Peter A. Stott, and Kirsten Zickfeld, "The proportionality of global warming to cumulative carbon emissions," Nature **459**, 829–832 (2009).

---

## Author Comment (AC7) · 25 Aug 2018

**CONTENTS**

**I. RESPONSES TO REFEREES AND CORRESPONDING REVISIONS**

In the following I have copied the comments made by the referees. Many thanks for their criticism, to which I have responded by incorporating numerous revisions leading to what seems to me to to a better and more balanced discussion. Detailed responses and the corresponding revisions are interpolated in the referee reports.

In most cases page and line numbers below refer to the revised version of the paper; follow this link. In some cases, as indicated, these numbers refer to the original version of this paper.

I also posted a version, retrievable via the discussion tab of this link, of the new version of the paper that shows the detailed changes. There are sufficiently many changes that this version is barely readable, but it does provide an impressionistic overview.

**A. Caldeira I**

This section contains responses to the referee comments found by following this link to RC1.

1   (a) **Caldeira I:** The title of this paper suggests that its primary function is not to act as a scientific paper but rather to support a normative claim.

      Concepts like what it means to be 'honest and responsible' in this context do not lend themselves to empirical tests, but rather express normative judgments.

GWPs are flawed metrics for almost every purpose, so I do not seek here to defend the use of GWPs. GWPs are metrics, and thus do not have a truth value. Like all tools, they can be more useful or less useful, but they cannot be right or wrong.

One could perhaps rephrase the title as: "Among global warming potential definitions, only the instantaneous global warming potential is useful", but this too is a matter of degree and depends on what you want to know.

(b) **Response:** The revised paper has a new title: *Proposed policymaker- friendly metric of radiative effects of greenhouse gases.* This should be a major step toward addressing the issues raised in these comments.

As far as values are concerned and statements that cannot be objectively confirmed, *i.e.*, are not strictly verifiable or falsifiable, this comes with the territory of policy tools and the trade-offs they necessarily imply. Whether or not a tool is useful depends on some sort of underlying utility function, which in turn will contain a scheme weighting choices and consequences, value judgments in other words.

The revised version of the paper avoids words such as "honest" and "responsible," which seem to have come across as inflammatory. They have been replaced by more neutral terminology. In addition, to address the confirmability issue, the new version of the paper features a more extensive discussion of and detailed references to IPCC's statement of purpose and criticism of the GWP going back to it Second Assessment; see page 2 paragraph at 20 and specific page references in footnote 1 on the same page.

Also included to provide context is a reference to a paper the title of which is *"Unmask temporal trade-offs in climate policy debates;"* see Ref. 1 in Science. All of this makes it clear that the proposed decision making tool contained in this paper is part of an ongoing scientific discussion at the interface between science and policy making.

2 (a) **Caldeira I:** If the author could come up with a cogent argument for what would be better than GWPs as conventionally defined that have not already been discussed in the literature, I would be open to reviewing that as a perspective or

opinion piece.

Addressing the normative claims, instantaneous radiative forcing values would seem to be a flawed metric for greenhouse gas accounting and attribution. Imagine two gases with the same instantaneous radiative forcing, but one decays in a year and the other remains in the atmosphere forever. Would it be wise to consider these two gases to be equivalent? The paper criticizes GWPs as conventionally defined but does not make a strong case for the use of instantaneous radiative forcing as an improvement. Indeed, many have criticized GWPs as conventionally defined for not considering effects on long time scales.

The author may want to resubmit as an opinion or perspective piece, but on a quick perusal I would not be enthusiastic to review that.

Ideally, in a policy context, one would like a metric to compare different greenhouse gases that would indicate the relative amount of damage that would be caused by an equal mass release of the different gases. This measure would be the ratio of the value of the damage caused by release of gas X to the damage caused by release of an equivalent quantity of CO2, where that time series of damage is appropriately turned into a scalar value to allow simple comparison.

Unfortunately, the estimation of future marginal damage and the conversion of time series to scalars (typically done in a net present value calculation) are rife with problems that have been widely discussed already. Further, the relative damage would also depend on the assumed background scenario against which these emissions occur. GWPs are used mostly because of historical legacy. They are clearly flawed metrics. Some people use them and are unaware with their deficiencies. Others use them, aware of their deficiencies. There are no doubt dishonest and irresponsible people who use these metrics to try to achieve nefarious ends. But users of flawed GWP metrics can be both honest and responsible.

(b) **Response:** As mentioned, the paper is presented as an attempt to construct a tool for policy makers. Indeed, the IPCC for the longest time has done exactly that and came up with the global warming potential (GWP) as a single number to quantify the joint effect of several greenhouse gases. The paper presents arguments to show that the time has come for science to provide a better tool than

the UNFCCC-blessed, widely used GWP with a 100-year horizon.

To substantiate this claim I expanded the discussion about outpaced climate change projections and "erring on the side of least drama" by including more context and additional references, such as Ref 2 and 3. There also is a new reference on tipping points—Ref. 4 in addition to *e.g.* the Hansen Ref. 5, which had already been included; see lines 3–13 on page 2.

The paper states explicitly that coming up with a simple, user-friendly decision making tool for a complex system such as the earth's climate is intrinsically difficult, if not impossible; see in particular page 2, lines 25–28.

The case for treating $CO_2$ as permanent and $CH_4$ as decaying is made by referring to Ref. 6 in the paragraph starting on line 6 on page 6.

The abstract, introduction (Sections 1) and the conclusions (Section 4) have been changed to reflect this. To be more specific, I included new references; see *e.g.* Refs. 7 and 8 and to references to recent, 2018 papers that highlight the decision timeframe and possible implications: see page 2 paragraph containing line 15.

(a) **Caldeira I;** And scientific papers should report previously unknown empirical facts, not value judgments.

(b) **Response:** If this sweeping statement were correct, reviews would not be scientific papers, nor would a vast segment of the applied mathematics literature qualify as scientific. Many of my publications in computational physics would not qualify. More specifically, and more importantly in the context of this exchange, it would also be impossible for scientific journals to contribute to providing better alternatives for the UNFCCC-accepted 100-year GWP. The view expressed by the referee would also disqualify Ref. 1 in Science and many other such papers. However that may be, as argued above, I rewrote the paper to be more compatible with what I think the referee really had in mind with this comment.

**B.   Caldeira II**

The following responses refer to referee comments posted in this link, RC2.

3  (a) **Caldeira II:** One way of framing this paper as a scientific paper would be to support the claim: "Different emissions scenarios that are equivalent on conventional GWP metrics produce very different climate outcomes." I am not sure how many papers, if any, already make that point compellingly. A more useful paper would be to provide a new metric such that different emission scenarios that are equivalent on this new metric would all have very similar climate outcomes.

The focus on instantaneous effects suggested in the title of the Nightingale's would not satisfy this objective.

(b) **Response;** The proposed tool produces a time-dependent metric, the area (as a function of time) under the curves shown in Figs. 3 and 4 and the curves in Figs. 6 and 7. These numbers are roughly proportional to the heat absorbed in the climate system over time by the radiative imbalance. Certainly, as far $CO_2$ is concerned that is what the referee is looking for. This is why I incorporated the citation to the results of Ref. 6. They are summed up in the paragraph starting on line 6 of page 6. Adding to that the integrated instantaneous effect of methane yields a reasonable approach to produce a heuristic measure to track the outcome of different emission scenarios.

The presence of "instantaneous" in the old title did not contradict any of this, because the actual tool features the time-integral of the instantaneous effect, *i.e.*, the cumulative effect. Reversely, by kicking the can down the road, as use of $GWP_{100}$ does, one will suppress the effect of different emission scenarios on a decadal time scale, a point already made in the previous version of the paper.

The citations to the work of Refs. 7 and 8 were added to provide further context for this line of reasoning. Are these arguments rigorous? No, not in the least, but they seem plausible, as argued both in the introduction of the paper and in the items listed on page 4, of which the first one (lines 1–3) is new.

**C.   Anonymous**

This section contains responses to the referee comments posted in this link, RC3.

4  (a) **Anonymous:** I recommend that this paper is rejected.

The study is well motivated but flawed. I had expected (from the abstract) to find some coherent reason why the instantaneous GWP is superior to the normal GWP(100). However, all I find (p 3;l 18-19) is an assertion that this IS the case and then the rest of the paper follows as if that assertion is justified. In fact the abstract contains no useful information about the content of paper, but only really states the assertion.

(b) **Response:** The main scientific reason why the 100-year horizon lacks justification was explained in the original version of this paper: it is the mismatch of timescales mentioned in lines 8–15 of page 3 of the original paper; these lines are currently in slightly expanded form—see next paragraph for details—present on line 27 of page 3 through line 8 of page 4. Also, there is the mathematical argument making $GWP_{100}$ unsuitable for dynamical tracking emssions time. This is mentioned on lines 18 and 19 of page 3 of the original paper, which correspond to lines 10–12 of page 4 in the current version.

Also the revised abstract makes reference to much of this; see lines 4 and 7 of page 1. In addition, the timescale mismatch argument has been strengthened in lines 14–16 of page 2, and lines 1–3 of page 4, as mentioned in the response in item 3.(b), and the newly incorporated references, *viz.* Refs. 7 and 8 on page 2 and Ref. 4 on page 4.

5 (a) **Anonymous:** I am no great fan of the GWP and the difficulties of using it to represent temperature change have long been known (its equivalence is formally restricted to time-integrated radiative forcing following a pulse emission). See for example Figure 3 of Fuglestvedt et al. (Climatic Change 58, 267-331, 2003) and many of the figures and references in Myhre et al. (2013).

There is much I disagree with in this paper, but I restrict myself to those aspects that I feel justify the rejection. The principal problem is that no account is taken of the much greater persistence time of CO2 perturbations, especially the fact that some of that CO2 is an essentially permanent addition to the atmosphere. This is acknowledged at p 2;l 28-29, but plays no subsequent part in the analysis. The only timescale used in the paper is methane's decay time.

(b) **Response:** As explained in detail in my response AR3, the principal problem

identified by the referee is based on a misconception. To clarify matters, the revised paper contains a more elaborate explanation of the approach. More explicitly, the fact that the equations feature two greenhouse gasses—one with an an infinite decay time, namely $CO_@$, and the other on $CH_4$, with a finite decay time—is explained more carefully in lines 6–8 on page 6 and also in the comments following Eq. (10), *i.e.*, in lines 15–18 on the same page. The rationale for the approach is also explained more carefully in the the newly added Ref. 6 on line 8 of page 6, which contains the justification of the treatment of the decay time of $CO_2$ as infinite in the proposed tool.

6 (a) **Anonymous:** The problem with the key figures (Figs 3 and 4) is that they just demonstrate the result of applying the assertion, rather than demonstrating that the assertion leads to a better representation of the resulting climate change than applying GWP(100), which is surely what matters. If the temperature effects (a simple physical model could be used in an illustrative context)) of using CO2-equivalents calculated using the GWP(0) was adopted, and compared with that resulting from the actual emissions (in the author's thought experiment) the temperature evolution of actual and CO2-equivalent emissions would be quite different. The impact of methane emissions from any given year would decay to near zero in a few decades, while much of its (large) equivalent in terms of CO2 using GWP(0) would remain in the atmosphere influencing climate for long periods.

(b) **Response:** I cannot follow what exactly the referee is driving at and do not know how to respond other than to say that part of what the referee writes seems to conflict with the contents of Figs. 1 and 2, but most of it might be related to the misconception addressed in the previous response, item 5.(b). As a matter of fact, the Eq. (10) describes exactly what the referee expects. I suspect that the revised version of the paper and in particular the new title explain more clearly what the purpose of the exercise is, namely to provide a better decision making tool than one of current UNFCCC scantioned, general use.

7 (a) **Anonymous:** The author invokes the precautionary principle but this only applies if the chosen metrics have demonstrable integrity. By placing a very large

multiplier on CH4 emissions, it would encourage large cuts to methane emissions in preference to those of CO2, but the longer-term consequences of such a choice would have to be explored to assess the extent to which such a policy is precautionary or ultimately leads to a greater climate change (which could only be reversed by the negative emissions that the author (p 9;l 7) regards as "fraught with danger").

(b) **Response:** As to the Precautionary Principle, its value as a public policy tool is not dependent on the proposed tool. The dependence runs in thee reverse direction. This, once again, is related to the timescale issue mentioned in item 4.(b). Given the vital importance of the decadal timescale, use of the 100-year time horizon is incompatible with the Precautionary Principle, as stated on line 32 of page 2, on line 8 of page 4, on line 29 of page 11, and one final time in line 2 of page 12. The revised version of the paper should do a better job of addressing this issue; indeed there are 7 references to the precautionary approach in the current version compared to the 5 in the original. The newly added Ref. 9 may help to convey that the Precautionary Principle is an accpeted ingredient of international treaties. It is therefore, the supreme law of the land. according to Article VI of the U.S. Constitution.

The sentence containing "fraught with danger" is no longer present; "negative emissions" is now part of a new first paragraph in the Conclusions section, lines 4–8 of page 11.

8 (a) **Anonymous:** The discussion surrounding Figures 1 and 2 is confused – again we are left with an assertion that the similarity between the Figures show consistency, when such consistency can only be demonstrated by converting emissions to changes in concentrations, changes in concentrations to radiative forcings and (transient) radiative forcing to (transient) changes in temperature. To do otherwise is to ignore the physics of the climate system. In essence, the attribution statements in IPCC AR5 are tracing through those necessary links.

(b) **Response:** If the climate system is perturbed by $CO_2$ emissions, a simple perturbation argument suggests that the changes in the temperature anomaly will develop with a delay along a similar trajectory. In other words, if one can fit

the emissions with an simple exponential curve, one would expect that the same exponent—hence the use of the term linear regression on line 7 of page 7—would describe the time dependence of the temperature anomaly. That is indeed the contents of lines 6 and 7 on that page. Maybe, if the goal were explicit prediction of the various proportionality constants and delays, one might have to follow the line of reasoning the referee seems to suggest. That would be an interesting calculation, but not one relevant within the scope of this paper.

I have to admit that it is not completely clear to me what the referee is looking for, but I expect that the newly included comments drawn from Ref. 6, mentioned above in the penultimate paragraph of items 2.(b) and in the first paragraph of item 3.(b), help explain the rationale for Figs. 1 and 2.

Let me also note that, in contrast to what the referee seems to suggest, the paper makes not attempt to provide an improved version of the Global Temperature change Potential (GTP) for example discussed on page 663 of Ref. 10. This might be an interesting exercise, but it is beyond the scope of the paper nor does it fit with the current UNFCCC practice of using the $GWP_{100}$.

**II. CLOSING COMMENTS**

Upon submission of the fist draft of the paper to Earth System Dynamics I conveyed to the editor that this was an unusual paper and not one that is typically found in science journals, my own publication record included.

Humanity is part of the dynamics of the earth and has brought about the Anthropocene. Prt of that dynamic is the fact is that the $GWP_{100}$ is used as a decision making tool. This is true in spite of the fact that, as Caldeira stated that GWPs "are flawed metrics for almost every purpose"—see item 1.(a)—while the anonymous referee mentioned to be "no great fan of the GWP"—see item 5.(a). It is indeed hard to think of a serious climate model that would use the $GWP_{100}$ as an input parameter. The problem that IPCC has tried to address by introducing the GWP as an decision making tool is that it it equally true that even the simplest climate models are too complicated for most policy makers.

This is the conundrum we face and all of this strongly suggests, to me at least, that it is important for the scientific community to bridge this gap and deal with the issues covered

in the paper. FOr that reason I do indeed greatly appreciate the editor's decision to put the paper out there for discussion.

I chose the original title of the paper—to a lesser extent the same applies to its contents—to convey that it contained both values and science. I seem to have gone overboard, bur in responding with changes to the referees' comments, I have tried to redress this problem.

———————————————

[1] I. B. Ocko, Steven P. Hamburg, Daniel J. Jacob, David W. Keith, Nathaniel O. Keohane, Michael Oppenheimer, Joseph D. Roy-Mayhew, Daniel P. Schrag, and Stephen W. Pacala, "Unmask temporal trade-offs in climate policy debates," Science **356**, 492–493 (2017).

[2] J. M. Melillo, T. C. Richmond, and G. W. Yohe, eds., *Climate Change Impacts in the United States: The Third National Climate Assessment* (U.S. Government Printing Office, 2014) pp. 1–841.

[3] Patrick T. Brown and Ken Caldeira, "Greater future global warming inferred from earth's recent energy budget," Nature **552**, 45–50 (2017).

[4] Sybren Drijfhout, Sebastian Bathiany, Claudie Beaulieu, Victor Brovkin, Martin Claussen, Chris Huntingford, Marten Scheffer, Giovanni Sgubin, and Didier Swingedouw, "Catalogue of abrupt shifts in intergovernmental panel on climate change climate models," Proceedings of the National Academy of Sciences , E5777–E5786 (2015), http://www.pnas.org/content/112/43/E5777.full.pdf?with-ds=yes.

[5] J. Hansen, "A slippery slope: how much global warming constitutes 'dangerous anthropogenic interference'?" Climatic Change **68**, 269–279 (2005).

[6] H. Damon Matthews, Nathan P. Gillett, Peter A. Stott, and Kirsten Zickfeld, "The proportionality of global warming to cumulative carbon emissions," Nature **459**, 829–832 (2009).

[7] Will Steffen, Johan Rockström, Katherine Richardson, Timothy M. Lenton, Carl Folke, Diana Liverman, Colin P. Summerhayes, Anthony D. Barnosky, Sarah E. Cornell, Michel Crucifix, Jonathan F. Donges, Ingo Fetzer, Steven J. Lade, Marten Scheffer, Ricarda Winkelmann, and Hans Joachim Schellnhuber, "Trajectories of the earth system in the anthropocene," Proceedings of the National Academy of Sciences (2018), 10.1073/pnas.1810141115, http://www.pnas.org/content/early/2018/07/31/1810141115.full.pdf.

[8] S R Rintoul, S L Chown, R M DeConto, M H England, H A Fricker, V Masson-Delmotte,

T R Naish, M J Siegert, and J C Xavier, "Choosing the future of antarctica," Nature **558**, 233–241 (2018).

[9] National Oceanic and Atmospheric Administration—Office of General Counsel, "Precautionary approach," (2017).

[10] T. F. Stocker, D. Qin, G.-K. Plattner, M. Tignor, S. K. Allen, J. Boschung, A. Nauels, Y. Xia, V. Bex, and P. M. Midgley, eds., *Climate Change 2013: The Physical Science Basis. Contribution of Working Group I to the Fifth Assessment Report of the Intergovernmental Panel on Climate Change* (Cambridge University Press, Cambridge, United Kingdom and New York, NY, USA, 2013) pp. 1–1535.